# Selective Phosphorization Boosting High-Performance NiO/Ni_2_Co_4_P_3_ Microspheres as Anode Materials for Lithium Ion Batteries

**DOI:** 10.3390/ma14010024

**Published:** 2020-12-23

**Authors:** Ji Yan, Xin-Bo Chang, Xiao-Kai Ma, Heng Wang, Yong Zhang, Ke-Zheng Gao, Hirofumi Yoshikawa, Li-Zhen Wang

**Affiliations:** 1School of Materials and Chemical Engineering, Zhengzhou University of Light Industry, Zhengzhou 450001, China; cxb@zzuli.edu.cn (X.-B.C.); mxk@zzuli.edu.cn (X.-K.M.); hengwangsci@163.com (H.W.); zy@zzuli.edu.cn (Y.Z.); gaokezheng@126.com (K.-Z.G.); 2School of Science and Technology, Kwansei Gakuin University, 2-1 Gakuen, Sanda, Hyogo 669-1337, Japan

**Keywords:** selective phosphorization, nickel cobalt, anode materials, lithium-ion batteries

## Abstract

Phosphorization of metal oxides/hydoxides to promote electronic conductivity as a promising strategy has attracted enormous attention for improving the electrochemical properties of anode material in lithium ion batteries. For this article, selective phosphorization from NiCo_2_O_4_ to NiO/Ni_2_Co_4_P_3_ microspheres was realized as an efficient route to enhance the electrochemical lithium storage properties of bimetal Ni-Co based anode materials. The results show that varying phosphorizaed reagent amount can significantly affect the transformation of crystalline structure from NiCo_2_O_4_ to intermediate NiO, hybrid NiO/Ni_2_Co_4_P_3_, and, finally, to Ni_2_Co_4_P_3_, during which alterated sphere morphology, shifted surface valance, and enhanced lithium-ion storage behavior are detected. The optimized phosphorization with 1:3 reagent mass ratio can maintain the spherical architecture, hold hybrid crystal structure, and improve the reversibly electrochemical lithium-ion storage properties. A specific capacity of 415 mAh g^−1^ is achieved at 100 mA g^−1^ specific current and maintains at 106 mAh g^−1^ when the specific current increases to 5000 mA g^−1^. Even after 200 cycles at 500 mA g^−1^, the optimized electrode still delivers 224 mAh g^−1^ of specific capacity, exhibiting desirable cycling stability. We believe that understanding of such selective phosphorization can further evoke a particular research enthusiasm for anode materials in lithium ion battery with high performances.

## 1. Introduction

Commercial lithium ion battery has been developed for almost 30 years, since it was first proposed by Sony corporation in 1991 [1]. Today, the fast development of electric vehicle has proposed a higher requirement for the lithium ion battery to possessing superior energy density, rapid charge/discharge rate, excellent cycling life, and environmental friendliness. As we know, graphite, as the second-generation anode material, has systematically been studied due to its advantages of low cost, cycling stability, and environmental benignity [2,3]. However, the low theoretical capacity of graphite (372 mAh g^−1^) has severely hindered its further application in electric vehicles, and this needs to be further improved or we will have to find new replaced anode materials to boost the development of advanced lithium-ion batteries [4]. In past decades, tremendous anode candidates have been proposed to remedy the deficiencies of commercial graphite [5]. Transition metal oxides (TMOs), as one of the most optimal candidates, have been researched extensively due to their higher electronic conductivity, relatively low cost, and reasonable cycling span, in comparison to their counterpart single metal oxides [6,7,8]. As alternative materials in lithium ion batteries, TMOs also show exceptionally specific lithium storage properties beyond 400 mAh g^−1^, higher than that of graphite [9]. Moreover, the low lithiation/delithiation potential, as well as cheap raw material, is believed to help in facilitating the further development of TMOs in lithium-ion batteries [10,11].

Phosphorization, as another efficient route to further enhance the electronic conductivity of NiCo_2_O_4_, has been successfully applied in supercapacitors [12], lithium-sulfur batteries [13], and electrocatalysts [14], as well as other research fields. Unfortunately, the large volume alteration and cycling capacity fading are still two impeding factors for potentially large-scale application of transition metal phosphides (TMPs) in lithium ion batteries [15]. For the first factor, the large volume alteration rises to the pulverization and particle aggregation during lithium-ion insertion process, especially in the initial several cycles. For the second factor, the low intrinsically electronic conductivity leads to the low utilization ratio of electroactive component, which tends to the limitation of commercial application [16]. The synergistic preparation of TMPs with carbon framework has been proven as an efficient approach to overcome both the drawbacks of volume expansion and low electronic conductivity. In our previous work [17], we have used solvothermal route to prepare NiCo_2_O_4_ and investigated its electrical conductivity and ion transfer behavior with different micro-structures in lithium ion batteries. During the investigation, we found that the microstructure severely affected the lithium storage properties. This phenomenon has triggered researchers to further consider the effect of selective phosphorization on microstructure, as well as electrochemical lithium storage, properties of NiCo_2_O_4_.

For this article, the influence of varying phosphorized reagent amount on crystal structure, morphological alternation, and the electrochemical properties of NiCo_2_O_4_ was systemically investigated. Interestingly, we found that the crystal structure exhibits significant alteration and spherical morphology was changed, as well. Owing to the detected phenomenon, the obtained hybrid NiO/Ni_2_Co_4_P_3_ microsphere electrode exhibits an improved lithium-ion storage capacity of 333.5 mAh g^−1^ at 100 mA g^−1^ specific current and manifests excellent rate capability at 5000 mA g^−1^, as well as high rate recover ability and outstanding cycling stability. The possible transformation mechanism from NiCo_2_O_4_ to NiO/Ni_2_Co_4_P_3_ and, finally, to Ni_2_Co_4_P_3_ is also discussed in detail.

## 2. Experimental

All chemicals used in this experimental section were purchased from Aladdin Corporation (Shanghai, China) and used as received without any further treatment or purification. The synthesis route can be divided into two steps: (1) the preparation procedure of NiCo_2_O_4_; and (2) the transformation from NiCo_2_O_4_ to Ni_2_Co_4_P_3_. For the preparation of NiCo_2_O_4_, the synthesis procedure keeps consistent with previous published literature [18]. In a typical route, 0.25 mmol Ni(NO_3_)_2_•6H_2_O and 0.5 mmol Co(NO_3_)_2_•6H_2_O was added a mixture solution of 16 mL glycerol and 80 mL isopropanol under strongly magnetic stirring. After the formation of pink transparent solution, the obtained mixture was transferred into 100 mL Teflon-lined stainless steel autoclave, sealed, and heated in an oven at 180 °C for 6 h. Finally, the cooled autoclave was unsealed, and the obtained precipitate was centrifuged, washed with ethanol several times to remove the residual raw materials, and dried in a vacuum oven at 80 °C for 12 h. In order to enhance the crystallinity, the dried precursor was sintered at 400 °C for 2 h at a ramp rate of 2 °C min^−1^ in air atmosphere. For the transformation from NiCo_2_O_4_ to Ni_2_Co_4_P_3_, the phosphorized reagent NaH_2_PO_2_ was put in the upper stem of furnace tube, and the former prepared NiCo_2_O_4_ was put in the down stem of the tube with the gap distance of 5 cm, and the weight ratio between NiCo_2_O_4_ and NaH_2_PO_2_ varied from 1:1, 1:3, 1:5 to 1:10. The phosphorization temperature was set at 300 °C for 2 h with high pure N_2_ flowing gas (50 mL/min). After the reaction, the obtained material was washed with oxygen-removed deionized water several times to wash out the un-reacted NaH_2_PO_2_ following drying at vacuum oven at 80 °C for 12 h. In order to distinguish the samples, the obtained products with above mentioned 1:1, 1:3, 1:5, and 1:10 weigh ratio were labeled as P-1:1, P-1:3, P-1:5, and P-1:10, respectively.

Crystal structure of all materials was characterized on X-ray diffractometer (XRD, Rigku MiniFlex 600, Tokyo, Japan) by ranging 2θ angle from 20° to 60° at a scanning rate of 2° min^−1^ with Cu Ka radiation. Morphological observation was carried out on scanning electron microscopy (SEM, JEOL JSM-7100F, Tokyo, Japan), and the micro inner morphology was detected by transmission electron microscopy (TEM, JEOL JEM-2010FEF, Tokyo, Japan). The distortion and graphitization of carbon for all materials were detected by Raman spectra (Raman, XloRA LabRAM, Montpellier, France). Surface chemical composition and valance information were collected by X-ray photoelectron spectrometric (XPS, ESCALAB 250 Xi, Waltham, MA, USA).

Electrochemical performances of all materials were evaluated by using coin cell, where the lithium foil was used as both counter electrode and reference electrode and 1 M LiPF_6_ in a mixture of ethylene carbonate: dimethyl carbonate: diethyl carbonate = 1:1:1 (volume ratio) was treated as organic electrolyte. The working electrode was prepared by a mixture of activated material with super P and polyvinylidene fluoride in a weight ratio of 80:10:10. The mixture was dispersed in some N-methyl pyrrolidone solution and stirred with a magnetic stirring bar for 12 h at room temperature. Once finishing the stirring, the resultant slurry was coated on copper foil (9 μm) by a doctor blade with a thickness of 15 μm at a coating speed of 25 mm/s. Then, we put the coated copper foil into a vacuum oven at 80 °C for drying over 12 h. After that, the as-prepared electrode film was punched into circle plates with a diameter of 12 mm for assembling coin cells. The average active material on each circle plate is about 2.0 ± 0.2 mg cm^−2^. The coin cells were assembled in a MB-Lab star glove box (M. Braun Coporation, München, Germany) with controlling the level of oxygen and moisture under 0.5 ppm. Celgard 2400 was used a separator with a diameter of 16 mm. Before electrochemical test, the assembled coin cells were set for 12 h to guarantee the well penetration of organic electrolyte into electrode. Galvanostatic charge/discharge tests were conducted on a multi-channel battery tester (LAND CT-2001, Wuhan, China) at various current densities from 3.0 to 0.01 V. Cyclic voltammetry was recorded on a CHI electrochemical station (CHI-660C, Shanghai Chenhua, China) from 3.0 to 0.01 V at a scanning rate of 0.1 mV s^−1^. Electrochemical impedance spectroscopy was measured on an Autolab electrochemical station (Autolab 302N, Herisau, Switzerland) in the frequency range of 10^5^ to 10^−1^ Hz with AC voltage of 5 mV.

## 3. Results and Discussion

After phosphorization, the crystal transformation from NiO, NiO/Ni_2_Co_4_P_3_ mixture, to Ni_2_Co_4_P_3_ has been summarized in Figure 1. When the ratio of NiCo_2_O_4_: NaH_2_PO_2_ reaches 1:1, the main diffraction peaks of P-1:1 sample can be ascribed to NiO (JCDPS 47-1049), which can be attributed to the low temperature phosphorization and low usage of NaH_2_PO_2_ [19]. It is interesting that there is no significant diffraction peak for both NiCo_2_O_4_ and Ni_2_Co_4_P_3_. As far as we know, the reason of disappeared NiCo_2_O_4_ is still unknown, and we are still working on it. Once increasing the ratio to 1:3, there is a mixture crystal phase of NiO and Ni_2_Co_4_P_3_. The existence of Ni_2_Co_4_P_3_ is believed to facilitate the improved electronic conductivity since it has higher electronic conductivity [13]. Further changing the usage of NaH_2_PO_2_ to five times that of NiCo_2_O_4_ (1:5), the peaks of NiO, almost weak or undetectable, and the crystal diffraction peaks of Ni_2_Co_4_P_3_ become stronger. When the ratio extended to 1:10, as shown in the pattern, there is no any detectable NiO peaks, and only diffraction peaks of Ni_2_Co_4_P_3_ can be observed. The alteration trend from NiO to Ni_2_Co_4_P_3_ certifies the strong influence of NaH_2_PO_2_ usage on crystal structure transformation.

The surface morphology of obtained products was checked by SEM, as shown in Figure 2. In comparison, we can see that the microsphere morphology can be well maintained, even as the ratio between NiCo_2_O_4_ and NaH_2_PO_2_ alters from 1:1, 1:3, to 1:5. It is concluded that the surface architecture is not changed from micrometer size observation. However, when the ratio jumps into 1:10, the microsphere has been severely destroyed, and there is no regular morphology observed, proving that the over usage of phosphatized reagent does not favor maintaining the unique sphere framework.

The inner architecture of P-1:3 sample (NiO/Ni_2_Co_4_P_3_) was further monitored by TEM. From Figure 3a–c, we can observe that the micrometer sphere is assembled by nano particles surrounded with visible pores, and its surface is relatively rough. There is an obvious gap between inner nanoparticles and pores, implying that it is not a unique solid sphere framework, as we previously reported [17]. This phenomenon can be ascribed to the influence of partly phosphorization. The expanded image in Figure 3d displays that there is significant separation between nanoparticles, and the particle boundary becomes rougher and vaguer, which further confirms the effect of partly phosphorization. Figure 3e further confirms the ratio of element, as well as the dispersion of nickel, cobalt, phosphorus and carbon. We can see that all elements are detected in the sample with the ratio of Ni, Co, and P reaching 2.12, 3.21, and 3.18. The good dispersion of carbon indicates the good electrical conductivity, which plays an important role in the stability of cycling performance.

Raman spectra of the as-prepared samples are shown in Figure 4. There are two significant band peaks detected from the image, where the G band at 1334 cm^−1^ stands for the graphization of the carbon, and the D band at 1580 cm^−1^ represents the disorder structure or the structural defects. It is noted that the relative intensity ratio between D band and G band of P-1:1, P-1:3, P-1:5, and P-10 (I_D_/I_G_) varies from 1.01, 1.03, 1.08, to 1.00. With the increase of phosphatized reagent usage, the peak intensity becomes stronger and then turns to weak again, indicating that the electronic conductivity of the electrochemical reaction has been promoted, and possible existed oxygen-functional group can increase the structural stability. The weak broadening of the G and D bands without significant change of I_D_/I_G_ ratio confirms the more disordered of surface phosphorized layer [20].

X-ray photoelectron spectroscopy (XPS) characterization was implemented to investigate the surface composition and valence change of Ni-Co-P samples. Figure 5a gives the overall XPS survey spectra, confirming the existence of Ni, Co, P, O, and C elements in the Ni-Co-P samples. High-resolution spectrum of Ni 2p, shown in Figure 5b, presents two distinct peaks at 855.8 and 873.6 eV, corresponding to the 2p_3/2_ and 2p_1/2_ and confirming the co-existence of Ni^2+^ and Ni^3+^. With increasing the phospharized usage, the 2p_1/2_ peak shifts towards high binding energy, illustrating the stronger chemical bond between Ni-P and the improved electronic conductivity. Meanwhile, the detectable peak at 852.7 eV related to Ni-P is only observed in P-1:3 sample [21]. After that, the 2p_3/2_ continually shifts to high binding energy, which can be attributed to the formation of NiO-POx bond [22]. For the spectra of Co 2p (Figure 5c), the two peaks at 781 and 797.2 eV can be assigned to 2p_3/2_ and 2p_1/2_. It is noted that, similar as Ni, the peaks of 2p_3/2_ also shift toward high bonding energy with increasing the phosphorized reagent usage. Similar results were detected with the conclusion that the appearance or formation of Co-P and CoO-POx. Additionally, as confirmed in Figure 5d, the P2p spectra consisted of P-O bond and P^δ-^, which are located at 132.9 and 129.7 eV, respectively. These two broad peaks are attributed to the air exposure. The positive shift of the P-O bond further confirms the increased electronic density for P and the alteration of chemical states of Co in the Ni-Co-P samples. It is interesting that there is also a significantly positive shift of peak in O1s spectra with an increase in the ratio between NiCo_2_O_4_ and NaH_2_PO_2_ (Figure 5e).

Figure 6 depicts the electrochemical impedance spectra (EIS) of all samples. From the pattern (a), all samples exhibit a semi-circle and a straight-like linear, which represent signals for charge transfer resistance (R_ct_) at high frequency and ion diffusion resistance (R_D_) at low frequency, respectively. In comparison with other samples, P-1:10 demonstrates the largest R_ct_ value (132 Ω). It is mainly because of the deterioration of microsphere structure under excessive phosphatized reagent. Among the other three samples, the P-1:5 possesses a smaller R_ct_ value (35 Ω) than that of P-1:1 (53 Ω) and P-1:3 (57 Ω) sample. However, the straight-like linear displays a significant difference, and more vertical linear is detected for P-1:3 sample. Since the high rate property of electrode material is always determined by ion diffusion capability, the direct evidence has relationship with linear in EIS. It is expected that the P-1:3 sample could deliver better rate capability than the other three samples. Additionally, the EIS patterns of Ni-Co-O, graphite, and P-1:3 electrodes are provided to further prove the improved charge transfer resistance after phosphorization, as shown in Figure 6b. It is noted that the Rct of P-1:3 is relatively smaller than that of both Ni-Co-O and graphite, confirming the enhancement of electrochemical reaction reversibility.

Galvanostatic charge/discharge and cycling stability tests of all samples were carried out to investigate the electrochemical performance of samples. When tested in 0.01–3.0 V as anode material in lithium ion batteries (Figure 7a), a specific capacity of 597 mAh g^−1^ is achieved at 100 mA g^−1^ specific current, while the value can be retained at 349, 281, 217, 120 mAh g^−1^ when the specific current increases to 200, 500, 1000, 2000, and 5000 mA g^−1^, respectively. After reset, the specific current to 100 mA g^−1^, the P-1:3 electrode still delivers 186 mAh g^−1^. Compared with other samples, as shown in Figure 7b, the P-1:1 sample exhibits a relatively much lower specific capacity at low specific current (100 mA g^−1^), and the P-10 sample demonstrates relatively lower specific capacity at high specific current (5000 mA g^−1^). The main reason for the poor rate capability of P-1:1 and P-1:10 can be attributed the lack of phosphorization and the destruction of sphere architecture, respectively. Among four samples, the P-1:3 and P-1:5 exhibit almost the same rate of capability and rate recovery ability. However, when both electrodes were cycled at 500 mA g^−1^ (Figure 7c), the reversible capacity of P-1:3 reaches 395.2 mAh g^−1^ and maintains at 224 mAh g^−1^ after 200 cycles, while the P-1:5 possesses 371 mAh g^−1^ for the 1st cycle but only delivers 172 mAh g^−1^ over 200 cycles. The better cycling stability of P-1:3, rather than that of P-1:5, may be due to the combination of hybrid structure of NiO/Ni_2_Co_4_P_3_, as well as perfect holding of microsphere structure.

Cyclic voltammetry (CV) of P-1:3 sample was recorded at a scan rate of 0.1 mV s^−1^, and the result is shown in Figure 8. The initial cycle of the sample demonstrates an obviously charging/discharging plateau pair at around 1.15 V/0.61 V, which could be attributed to the decomposition of NiO/Ni_2_Co_4_P_3_ to metallic nickel and cobalt [16,23,24]. The whole discharging reaction (1) and corresponding charging reaction (2) are summarized as follows:(1)$$Ni_{2}Co_{4}P_{3}+3Li^{+}+3e^{-}↔Li_{3}P+4Co+2Ni,$$
(2)$$\begin{array}{l} 2Ni+Li_{3}P↔Ni_{2}P+3Li^{+}+3e^{-}\\ 2Co+Li_{3}P↔Co_{2}P+3Li^{+}+3e^{-} \end{array}$$

However, two reduction peaks located at 1.52 V and 1.08 V could be caused by the intercalation of lithium ion into the Ni_2_Co_4_P_3_ and the formation of Li_3_P and metallic cobalt and nickel, respectively. Another reduction peak at 0.61 V may be explained by the formation of solid electrolyte interphase (SEI) film. In the second cycle, the cathodic peak shifts towards higher voltage, while the anodic peak moves to lower voltage, which is explained by the phosphorization of metallic nickel and cobalt. The shift of two redox peaks from CV can be explained by the decomposition of SEI film, while the two reduction peaks at 1.01 V and 0.44 V are ascribed to the transformation of Ni^2+^ to Ni and Co^2+^ to Co, respectively. It is worth noting that the weak peak intensity indicates the irreversible capacity decay from the formation of SEI film.

The surface morphology of P-1:3 sample before/after test is shown in Figure 9. It can be seen that there is no significant sphere crack or morphology alteration after test. The microsphere architectures are well maintained, even as the sphere surface becomes relatively rough, which can be ascribed to the electrochemical reaction with lithium ion during the test. The well maintaining of the sphere framework further proves the excellent electrochemical cycling stability and superior rate capability.

## 4. Conclusions

In summary, the selective phosphorization of nickel cobalt oxides has been explored to improve the lithium ion storage properties under different usage of NaH_2_PO_2_. Under the optimized usage, the as-formed NiO/Ni_2_Co_4_P_3_ exhibits improved capability and cycling stability, which are 415 mAh g^−1^ at 100 mA g^−1^ and 224 mAh g^−1^ after 200 cycles at 500 mA g^−1^, respectively. The combination of maintaining micro-sphere architecture and partly phosphorization is believed to favor achieving such high lithium ion storage performances. Hybrid composites with oxides and phosphides are exhibiting promoted electrochemical energy storage performance and are one kind of the promising candidates to meet next-generation energy storage devices.

## Figures and Tables

**Figure 1 materials-14-00024-f001:**
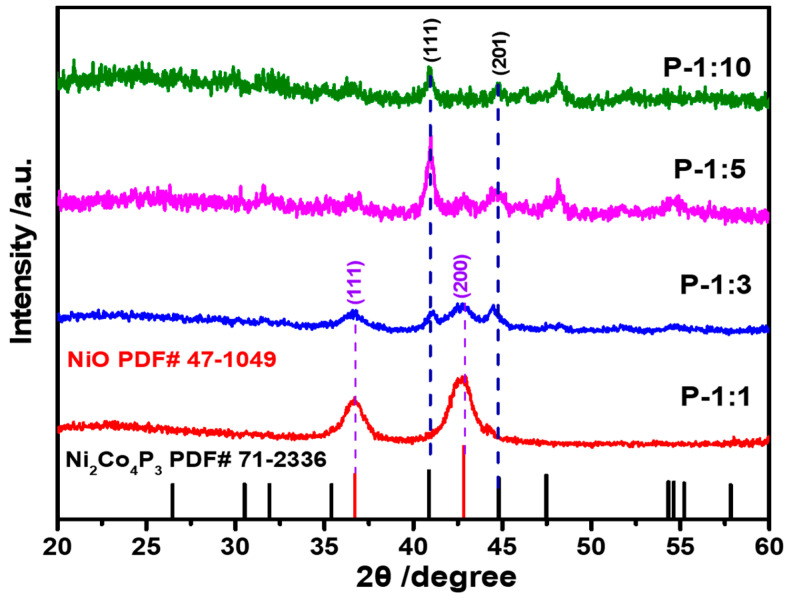
XRD pattern of Ni-Co-P samples with different ratio from 1:1, 1:3, 1:5, and 1:10.

**Figure 2 materials-14-00024-f002:**
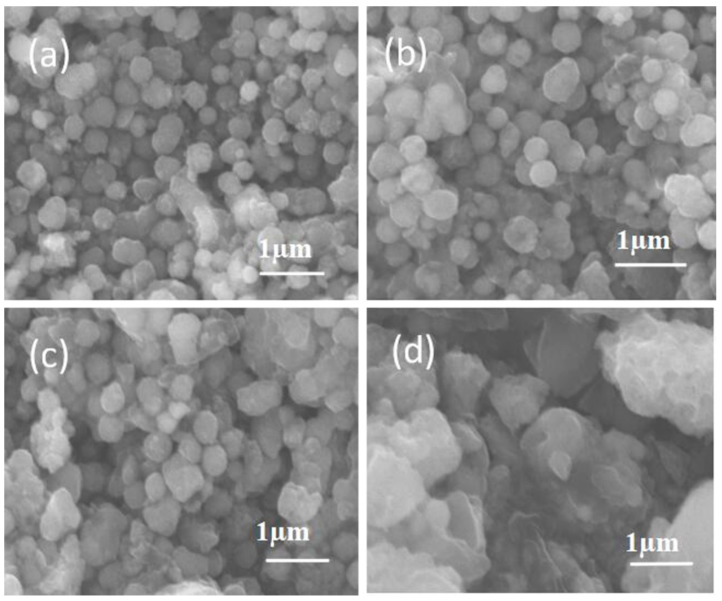
SEM images of Ni-Co-P samples with different ratio from (**a**) 1:1, (**b**) 1:3, (**c**) 1:5, and (**d**) 1:10.

**Figure 3 materials-14-00024-f003:**
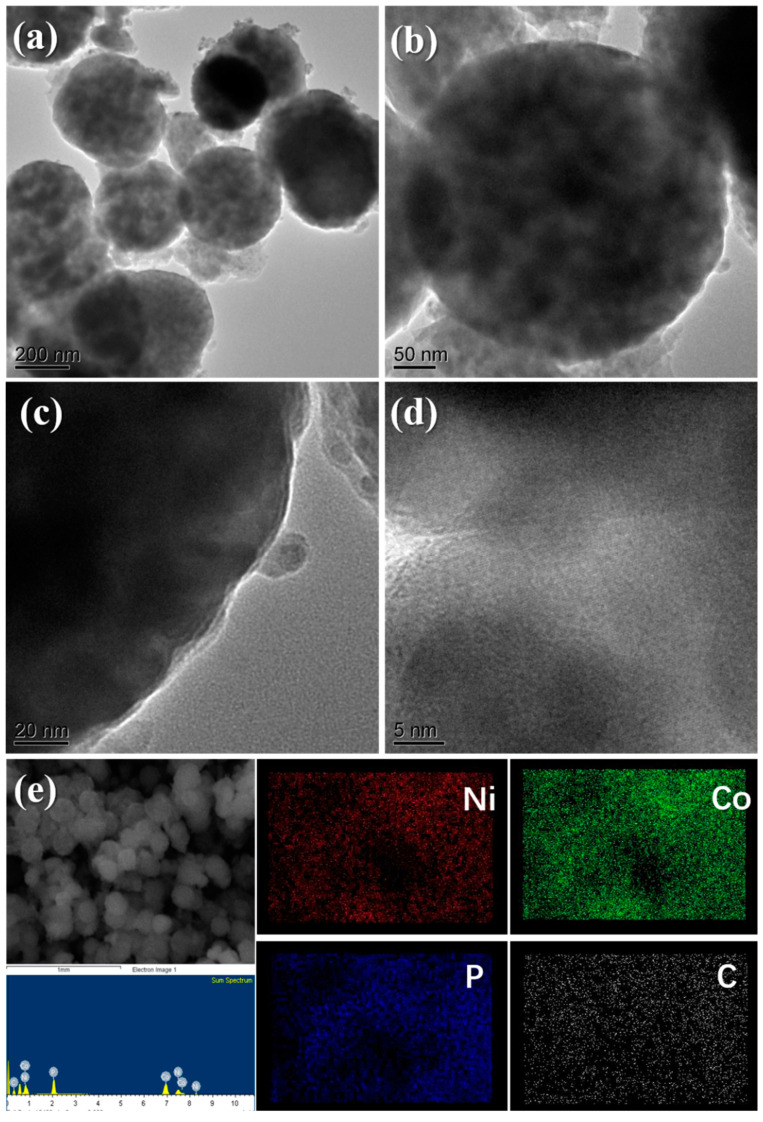
(**a**–**d**) TEM images of P-1:3 sample; (**e**) element mapping.

**Figure 4 materials-14-00024-f004:**
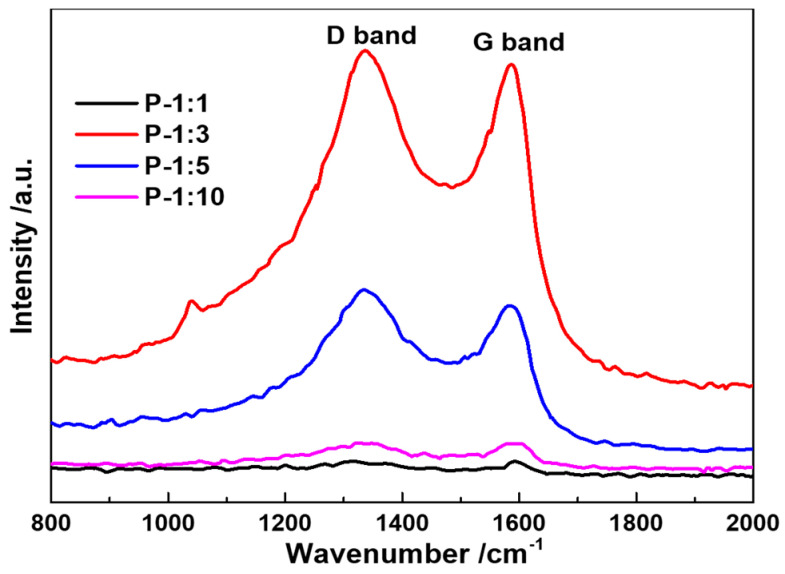
Raman spectra of Ni-Co-P samples with different ratio from 1:1, 1:3, 1:5, and 1:10.

**Figure 5 materials-14-00024-f005:**
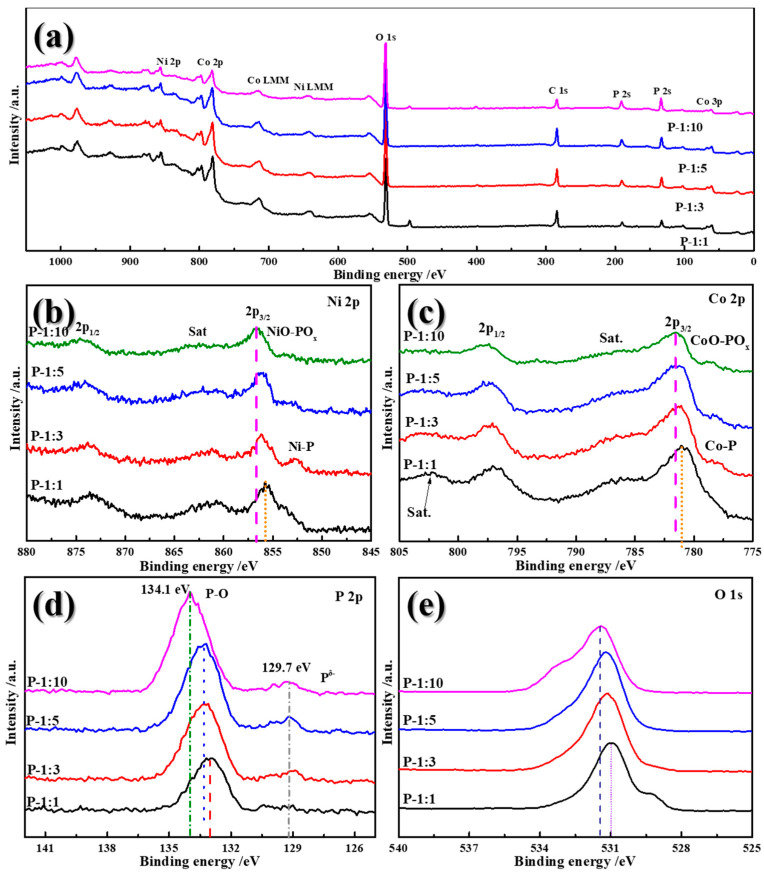
XPS spectra of Ni-Co-P samples with different ratio from (**a**) 1:1, (**b**) 1:3, (**c**) 1:5, and (**d**) 1:10, (**e**) O1s spectra.

**Figure 6 materials-14-00024-f006:**
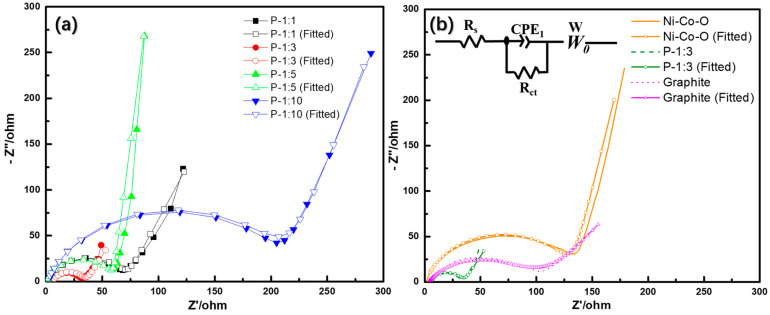
(**a**) Electrochemical impedance spectra (EIS) pattern of Ni-Co-P samples with different ratio from 1:1, 1:3, 1:5, to 1:10. (**b**) Comparison of EIS pattern between Ni-Co-O, P-1:3, and graphite.

**Figure 7 materials-14-00024-f007:**
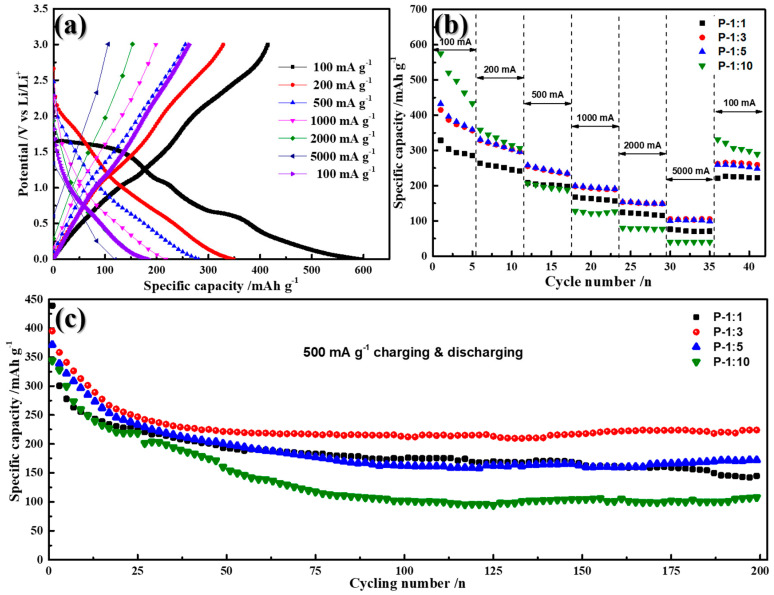
(**a**) Galvanostatic charge/discharge test GCD profiles of Ni-Co-P sample with 1:3 ratio; (**b**) rate capability of Ni-Co-P samples at various scanning rates; (**c**) cycling stability of Ni-Co-P samples with different ratio at a specific current of 500 mA g^−1^.

**Figure 8 materials-14-00024-f008:**
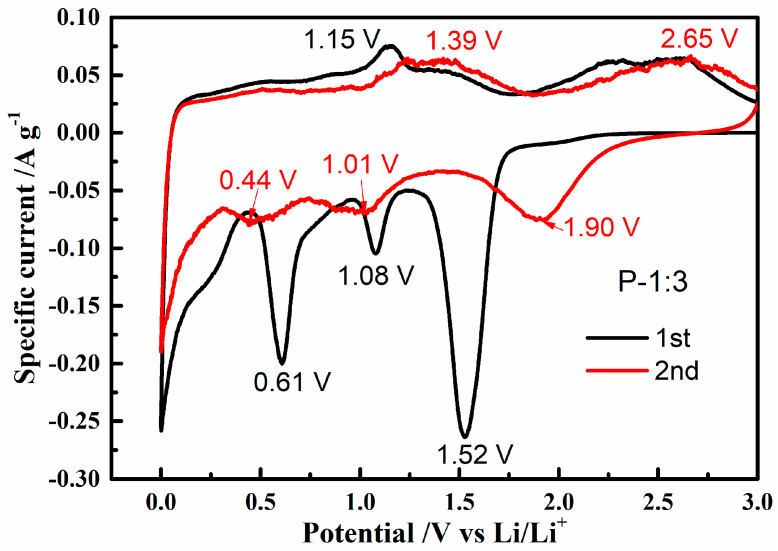
Cyclic voltammetry (CV) curves of Ni-Co-P sample with 1:3 ratio.

**Figure 9 materials-14-00024-f009:**
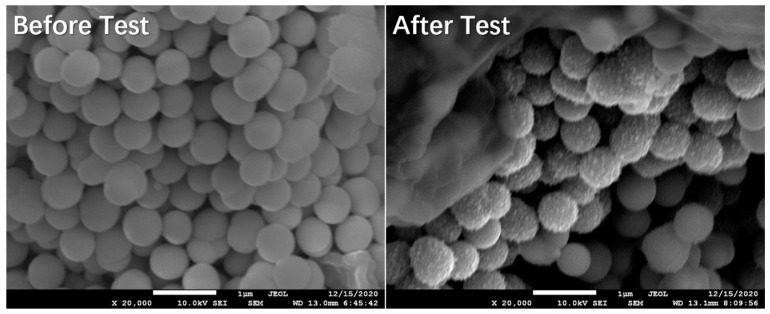
Surface morphology of P-1:3 sample before/after test.

## Data Availability

Data sharing is not applicable to this article.

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
