# Peer review of "Selective Phosphorization Boosting High-Performance NiO/Ni2Co4P3 Microspheres as Anode Materials for Lithium Ion Batteries"

_materials, 2020, doi:10.3390/ma14010024_

Round 1

Reviewer 1 Report

The paper is well written and explained. The topic presented is of the utmost interest for the next generation Li-ion batteries. The results are promising and well introduced by the research results. Only a few spell checks should be done by a second reading as for instance "Raman spectra spectrum" (page 3 line 96).

The list of compounds "EC:DMC:DEC" should be expressed without using the codified acronyms being the other compounds always presented with their official name.

The microstructural features of the electrode are supported by pictures however it would be very interesting to have an image fo the same materials after testing in order to better define the degradation rate (if any) of the electrode materials.

Fig. 3 is cut at 3V why? there is a visible continuation over such limit.

In the abstract and in the text it is claimed that the electrode has an intrinsic conductivity and EIS curves are shown however it would make more clear the presentation of such data if the EIS curves were accompanied by a table of data and an equivalent circuit so that the resistivity of the electrode is defined and compared with the one of graphite (on one side) and other similar but not phosphorized electrodes (on the other side).

Reading this paper is interesting and triggering the need to understand how the Li is intercalated into the electrode. Being an Anode we suppose that Li is intercalated as reduced atom, is it correct? The authors are kindy invited to discuss the intercalation mechanism to better clarify how this electrode works.

Author Response

Please see responses with figure and equations in attached PDF file.

Reviewer 1 comments:

  • The paper is well written and explained. The topic presented is of the utmost interest for the next generation Li-ion batteries. The results are promising and well introduced by the research results. Only a few spell checks should be done by a second reading as for instance "Raman spectra spectrum" (page 3 line 96).

Response: Thank you for your comments. We have carefully rechecked the English language used in the manuscript to avoid those low-level errors.

  • The list of compounds "EC:DMC:DEC" should be expressed without using the codified acronyms being the other compounds always presented with their official name.

Response: Thank you for your comments. We have provided the full name of used electrolyte organic reagent including EC is ethylene carbonate, while DMC and DEC are dimethyl carbonate and diethyl carbonate, respectively.

  • The microstructural features of the electrode are supported by pictures however it would be very interesting to have an image fo the same materials after testing in order to better define the degradation rate (if any) of the electrode materials.

Response: Thank you for your comments. We have provided the surface morphology of electrode after testing to further double check the maintaining of electroactive material. We can observe from the following images before/after test that there are few crack of microspheres after testing and the surface rough might cause by the reaction with lithium ion during cycling, proving the well holding of architecture after cycling test.

  • 3 is cut at 3V why? there is a visible continuation over such limit.

Response: Thank you for your comments. For anode material, if the charging limited voltage beyond 3.0V, it may result in the oxidation transformation from sulfide or phosphide to oxide, thus in this study, we adopted common limit voltage at 3.0V to avoid this side reaction.

  • In the abstract and in the text it is claimed that the electrode has an intrinsic conductivity and EIS curves are shown however it would make more clear the presentation of such data if the EIS curves were accompanied by a table of data and an equivalent circuit so that the resistivity of the electrode is defined and compared with the one of graphite (on one side) and other similar but not phosphorized electrodes (on the other side).

Response: Thank you for your comments. We have provided the tested results for graphite, phosphorized electrode and non phosphorized electrode as well as the corresponding equivalent circuit of EIS as you suggested. Simultaneously, since the purpose of this manuscript is to determine the influence of phosphorized extent on the structure and electrochemical activation of Ni-Co-based composites, we also stimulated the samples with different phosphorized extent in the following right image.

  • Reading this paper is interesting and triggering the need to understand how the Li is intercalated into the electrode. Being an Anode we suppose that Li is intercalated as reduced atom, is it correct? The authors are kindly invited to discuss the intercalation mechanism to better clarify how this electrode works.

Response: Thank you for your comments. Actually, the lithiation of Ni2Co4P3 can be explained from two aspects. The discharging reaction is ascribed to the transformation of Ni2Co4P3 to Li3P as well as the formation of metallic cobalt and nickel, while the charging reaction is attributed to the metallic reaction between nickel/cobalt and Li3P to form Ni2P/Co2P, as shown in following equation:

        Discharging reaction

         Charging reaction

Reviewer 2 Report

The paper provides interesting information with respect to Ni-based anode materials for Li ion batteries. Though, some points need to be revised.

Abstract:

“100 mA g-1” and “5000 mA g-1” is not a “current density”, it is specific current as it is relativated per mass. Please revise in whole manuscript.

Introduction

“As we know, graphite, as the first-generation anode material, has systematically been studied due to its advantages of low cost, cycling stability and environmental benignity[2].”

Amorphous carbon is the first generation anode material. Graphite corresponds to second generation. Please revise and include the literature:

Doi 10.1021/Cr030203g

  1. Winter, J.O. Besenhard, M.E. Spahr, P. Novak, Adv. Mater., 10 (1998) 725-763.

„However, the low theoretical capacity of 35 graphite (372 mAh g-1) has severely hindered its further application in electric vehicle, which 36 becomes the main bottleneck for boosting the development of advanced lithium-ion batteries.”

Actually, the bottleneck is the cathode, which reveals half the capacity of graphite, i.e. Ni rich NMC with ≈ 170 mAh g-1. The capacity of cathode is consequently the bottleneck, thus would have more impact on performance after improvement, though capacity of anode should be further enhanced, as well. Please revise and include the literature with the respect capacity calculations: https://doi.org/10.1007/s41061-018-0196-1.

Figure 6: Why EIS instead of the galvanostatic method, i.e. voltage hysteresis, IR-drop, etc. The latter would be less complex and at application near conditions. Please provide the reason for EIS. See for example the insights which can be obtained from simple overpotentials 

Figure 8: the y axis should be specific current.

Author Response

Reviewer 2 comments:

The paper provides interesting information with respect to Ni-based anode materials for Li ion batteries. Though, some points need to be revised.

  • Abstract: “100 mA g-1” and “5000 mA g-1” is not a “current density”, it is specific current as it is relativated per mass. Please revise in whole manuscript.

Response: Thank you for your comments. We have carefully revised the claim in whole manuscript.

  • Introduction: “As we know, graphite, as the first-generation anode material, has systematically been studied due to its advantages of low cost, cycling stability and environmental benignity[2].”

Amorphous carbon is the first generation anode material. Graphite corresponds to second generation. Please revise and include the literature: Doi 10.1021/Cr030203g  Winter, J.O. Besenhard, M.E. Spahr, P. Novak, Adv. Mater., 10 (1998) 725-763.„However, the low theoretical capacity of 35 graphite (372 mAh g-1) has severely hindered its further application in electric vehicle, which 36 becomes the main bottleneck for boosting the development of advanced lithium-ion batteries.”

Actually, the bottleneck is the cathode, which reveals half the capacity of graphite, i.e. Ni rich NMC with ≈ 170 mAh g-1. The capacity of cathode is consequently the bottleneck, thus would have more impact on performance after improvement, though capacity of anode should be further enhanced, as well. Please revise and include the literature with the respect capacity calculations: https://doi.org/10.1007/s41061-018-0196-1.

Response: Thank you for your comments. We have changed the claim to avoid the misunderstanding for the graphite as second generation anode material. Simultaneously, we totally agree with you that the bottleneck of lithium-ion battery performance is limited by cathode material rather than anode material. We have revised the sentence and the corresponding section in the revised manuscript and cited the literatures (Doi: 10.1021/Cr030203g; Doi: and Doi: 10.1007/s41061-018-0196-1) you recommended.

  • Figure 6: Why EIS instead of the galvanostatic method, i.e. voltage hysteresis, IR-drop, etc. The latter would be less complex and at application near conditions. Please provide the reason for EIS. See for example the insights which can be obtained from simple overpotentials 

Response: Thank you for your comments. The selection of EIS to identify the electrochemical process of lithium ion battery can be attributed to the following reasons: 1) the EIS is always used to detect the electrochemical behavior under the required status that the electrochemical reaction step is speed-control step rather than liquid phase diffusion step; 2) the EIS can be used to detect the diffusion coefficient of lithium ion at any selected charge/discharge situation (OCV, fully charge status or fully discharge status) while the calculation results of GITT or PITT is controlled by the charge/discharge current and relax time, which are determined by different people and the decision to justice whether the system has finished the electrochemical reaction may varies by different criterion standard. Thus, based on above consideration, the EIS has been selected in the study to provide the information of electrochemical progress.

  • Figure 8: the y axis should be specific current.

 Response: Thank you for your comments. As you suggested, we have corrected the y axis to specific current.

Round 2

Reviewer 2 Report

Good revision work of the authors. Can be accepted in present form.